# Giant Water Clusters: Where Are They From?

**DOI:** 10.3390/ijms20071582

**Published:** 2019-03-29

**Authors:** Tatiana Yakhno, Mikhail Drozdov, Vladimir Yakhno

**Affiliations:** 1Federal Research Center Institute of Applied Physics of the Russian Academy of Sciences (IAP RAS), Nizhny Novgorod 603950, Russia; 2N. I. Lobachevsky State University of Nizhny Novgorod (National Research University), Nizhny Novgorod 603950, Russia; 3Institute for Physics of Microstructures RAS (IPM RAS), Nizhny Novgorod 603087, Russia; director@ipmras.ru

**Keywords:** free and bound water, giant water clusters, liquid-crystal shells, salt microcrystals

## Abstract

A new mechanism for the formation and destruction of giant water clusters (ten to hundreds of micrometers) is proposed. Our earlier hypothesis was that the clusters are associates of liquid-crystal spheres (LCS), each of which is formed around a seed particle, a microcrystal of sodium chloride. In this study, we show that the ingress of LCSs into water from the surrounding air is highly likely. We followed the evolution of giant clusters during the evaporation of water. When a certain threshold of the ionic strength of a solution is exceeded, the LCSs begin to “melt”, passing into free water, and the salt crystals dissolve, ensuring re-growth of larger crystals as a precipitate on the substrate. A schematic diagram of the dynamics of phase transitions in water containing LCSs during evaporation is proposed. The results illustrate the salt dust cycle in nature.

## 1. Introduction

The structure and dynamics of water has been a major debatable topic for more than a decade [1,2,3,4,5]. To explain some of its anomalous thermodynamic properties (isobaric temperature dependence of density, thermal expansion coefficient, isothermal compressibility, and isobaric heat capacity) compared to simple liquids, a two-phase water concept has been developed [6,7]. The reasons for the anomalous physical properties of water are traditionally sought at the atomic-molecular level, in the nanometer space, and on the picosecond time scale. However, recent data have defined water as a micro-dispersed system. To the best of our knowledge, giant (up to millimeter-sized) clusters in a thin layer of water were detected for the first time using the IR (Infrared) spectroscopy [8] and it was suggested that they have liquid-crystal nature. Subsequently, supramolecular water complexes, ranging in size from 10 to 100 μm, were visualized using laser interferometry [9,10] and small-angle light scattering [11,12]. The nature of these clusters is still controversial. The authors of [13] used dielectrometry and the resonance method to show that with an increase in the frequency of reactive current from 1 to 300 kHz, the electrical capacitance of distilled water decreases several-fold. With increasing concentration of NaCl aqueous solutions, their electrical capacity also increases several-fold. It was suggested that these changes in distilled water were due to the presence of interconnected associates. Changes in NaCl solutions, according to [13], depend on the ratio of the number and size of water associates and the degree of hydration of ions. A decrease in the content of water aggregates (*r* ≤ 5 μm) under the action of an external permanent magnetic field with a magnetic induction of 1.5 T for 30 min was also noted in [14]. 

In our previous studies, using acoustic impedancemetry, we identified slow (near-hour) self-oscillation processes in colloidal liquids [15] and proposed a mechanism for their implementation [16]. It is based on periodic phase transitions between the free and bound (liquid-crystal) water of the hydration shells of the dispersed phase, which can be traced by fluctuations in the density of the liquid. These transitions are controlled and coordinated in the entire volume by osmotic pressure changing as a result of these events. Morphologically, these processes manifest themselves as the growth and destruction of spherical structures of micron size (50–250 μm) transparent in a liquid medium and visible in it due to the “contouring” by the particles of the dispersed phase. In-phase with the growth and destruction of water microstructures, the surface tension of the test solution fluctuates at the “liquid–air” interface. Liquid-crystal water spheres exhibit the properties of a viscous liquid and evaporate at temperatures above 200 °C [17]. 

The existence of a near-wall layer of water with special physical properties has been known for about 100 years [18,19,20]. A near-wall layer of water with unusual physical properties has also been described for colloids [21]. In recent years, the concept of an “exclusion zone” (EZ) has been developed [22,23]. EZ is a thin water layer (up to 500 µm) with a denser molecular packing than free water. This zone is formed during 10 to 60 min [22,23]. In the course of this formation, other ions and dispersed particles are displaced from the close-packed water zone. The growth of EZ near the surface of metals and glass was also noted [22,24]. We believe that LCSs observed by us are also EZs formed around salt microcrystals. The concept of the molecular structure of the EZ has evolved from multi-layered packing of water dipoles [23,25] to a layered structure of flat sheets of hexagonal cells formed by water molecules, where the sheets, unlike crystalline ice, are interconnected by weak electrostatic interaction rather than by rigid hydrogen contacts [26]. This ensures that the sheets glide relative to each other, as a result of which the EZ behaves like a viscous liquid. It is noteworthy that in accordance with the study [27], during the melting of ice, a liquid-crystal phase of water forms first and then free water. Based on the results of our work [15,16,17], we continued to study the origin and areas of existence of LCSs when considering the phase transformations of water in the process of its evaporation. The structure and atomic composition of materials were analyzed by the method of X-ray diffraction and secondary ion mass spectrometry (SIMS).

## 2. Results

In our earlier work we showed that under natural conditions, water was a dispersed system, in which the dispersed phase was represented by salt microcrystals surrounded by thick hydrated shells—liquid-crystal spheres [28]. LCSs were also present on the surface of dry glass and plastic. However, the origin of salt microcrystals in a liquid medium remained unknown. In the present study, we show that on the third day of natural evaporation of distilled water (specific electrical conductivity being 4.1 µS/cm) placed in a hydrophobic plastic Petri dish with a 2 mm thick layer, a thin film of non-evaporating water was observed (Figure 1a). The use of a polarization filter made it possible to detect the luminescence of liquid-crystal water shells around the central salt microcrystals (Figure 1b). When the focal length was changed in the same frame, the central salt microcrystals began to glow (Figure 1c). After free evaporation of a layer of distilled water ~2 mm thick from a glass Petri dish under room conditions, a loose sediment remained at the bottom, which eventually turned into an array of salt crystals (Figure 2). The electrical conductivity of distilled water used in this experiment was 36.5 µS/cm. 

Passing the air of the working area through the water sample with specific electrical conductivity 4.1 µS/cm for 10 min at a speed of 72 liters per hour led to an increase in this indicator. A week later, after the restoration of the water structure disturbed by air barbation, the electrical conductivity of the control sample was 4.1 µS/cm, and of the experimental one 5.4 µS/cm. The microscopy of the dried droplets of these water samples tested on the glass slide also revealed the difference in the content and structure of the sediment (Figure 3). The passage of air through the water sample led to severe water pollution. Moreover, salts constituted a large percentage of the microimpurities, which ensured an increase in the electrical conductivity of the sample. Thus, the aerosol origin of LCSs is quite likely. 

Analysis of the crystals using an X-ray diffractometer unambiguously confirmed that it was sodium chloride (Figure 4). That is, NaCl microcrystals were really “priming” for the formation of LCSs. 

Secondary ion mass spectrometry was used to measure static mass spectra from the surface of a clean silicon wafer before and after evaporation on it of a drop of deionized water stored in the laboratory in a closed glass vessel. The analysis showed that in the dried drop of water, a strong increase in the intensity of all lines containing carbon and sulfur was noted (Figure 5 and Figure 6). Moreover, the distributions of positive ions on the line of the three-phase boundary of the dried drop and inside it were different: in the area of the line the content of Li, Na, Mg and K increased (Figure 7). Of the negative ions on the line of the three-phase boundary, the chlorine content also increased (Figure 8).

The radial distribution of ions over the spot of the dried drop was uneven (Figure 7). In the central zone of the spot of the dried drop, the content of carbon, sulfur and chlorine prevailed, whereas lithium, sodium, potassium, magnesium, calcium and chlorine ions were more concentrated at the periphery. The presence of carbon and sulfur in a dried drop may indicate the presence of soot, which, like salt, could get into the water from the air.

We can make sure that the content of alkali metals and chlorine in the area of the perimeter of the drop is increased. Let us compare a similar map of the surface of a silicon substrate outside the zone of location of the dried drop and inside it (Figure 9).

The results reveal the presence of contaminations both on the surface of the substrate, and in the remains of the dried water. It can be verified that in the dried drop zone the intensity of several secondary ions, including Na and K, is higher than on the substrate. This suggests that these contaminants were present in the water. 

The dilution effect of small amounts of salts on colloidal structures has been actively discussed since the middle of the last century ([29], p. 50). Let us consider in more detail the evolution of structures on a glass slide when a portion of tap water (1 mL) containing microimpurities dries out (Figure 10). At the beginning of drying of the water sample, part of the LCSs, as a colloidal phase, is transferred by a capillary flow to the three-phase boundary [30,31] (Figure 9a,b). A progressive decrease in the concentration of free water is accompanied by an increase in the ionic strength of the solution. At the same time, part of the liquid-crystal water surrounding the salt “seed” under the action of osmotic pressure melts, turning into free water and creating conditions for the growth of crystals with further evaporation of free water.

After drying of 1 mL of distilled water with a specific electrical conductivity of 36.5 µS/cm, the very beginning of salt erosion of the LCS units at the interface was observed on a glass slide (Figure 11a,b). In the dried sample of water with a lower specific electrical conductivity (14.0 µS/cm), large aggregates of LCSs settling on the substrate were found (Figure 11c,d).

Experimental data and available literature sources presented in this work allowed us to draw a schematic diagram of the dynamics of phase transitions in water containing LCSs with NaCl microcrystals as a “seed” when it evaporates from a solid substrate (Figure 12).

## 3. Discussion

In the present study, we describe the nature of the giant clusters in water. Earlier studies showed that after evaporation of distilled water (specific conductivity—4.1 µS/cm) in natural laboratory conditions, a non-evaporating thin water film remained on the substrate containing inclusions in the form of spherical structures (LCSs) with a black dot in the center. The size and morphological features of these structures were similar to those observed earlier in a thin (~8 μm) fluid layer [28]. In polarized light with different focal lengths, it was possible to observe either the luminescence of liquid-crystal water shells or salt microcrystals inside them (Figure 1). After evaporation of water with a higher (by a factor of 9) electrical conductivity, the density of the sediment was much higher (Figure 2a). Two weeks later, crystal growth was noticeable (Figure 2b). This growth was significantly accelerated after merging the sediment into a single array (Figure 2c,d). The structure of the crystals corresponded to sodium chloride. This fact was also confirmed crystallographically (Figure 4). The transmission of 12 liters of air of the working laboratory area through 50 mL of distilled water led to an increase in its conductivity by 30%. At the same time, the total amount of impurities in water increased substantially (Figure 3). It is obvious that the salt component was also present in the composition of air pollution.

The study of deionized water droplets on a silicon substrate using SIMS showed the presence in this system of a rich set of chemical elements, including carbon and sulfur (Figure 5 and Figure 6). These elements are part of the soot—the usual component of the air of cities. During preparation for the study, both the water and the substrate had direct contact with the air. However, the carbon and sulfur content in the dried water residues was higher than on the substrate. We suggested that the source of soot penetration into deionized water could be its contact with air.

It is known that in the process of drying of a drop of aqueous solutions on a solid wetted surface, thermocapillary flows arise in a liquid, leading to the redistribution of components and the formation of an edge ring along the three-phase boundary of this drop [30]. In our experiment, after the drop was placed on a silicon substrate, it began to rapidly dry in a dry-heat cabinet. As a result, it was possible to observe only the beginning of the formation of an edge ring and the disordered deposition of impurities contained in water over the entire drop area (Figure 7 and Figure 8). Crystallization of salts (chlorides) of sodium and potassium at the three-phase boundary of the drying drop is a regular phenomenon [31].

Thus, the results of the work indicate that the composition of air dust includes salt components (mainly sodium chloride) [32]. Falling into the water, they become the seeds of “giant clusters” (LCSs) and form a dispersed phase of water. When a certain volume of water with high electrical conductivity evaporates on a solid substrate, LCSs undergo a cascade of phase transitions—from melting LCSs and merging drops of salt solutions (Figure 8) to subsequent recrystallization with the formation of larger crystals. Figure 9a,b shows the beginning of the process of destruction of LCSs during water evaporation and an increase in the ionic strength of a solution with a lower initial conductivity and (c,d) aggregation of LCSs in drying water.

The authors’ views on the dynamics of phase transitions in water containing LCSs are reflected in a schematic diagram (Figure 12). 

According to NASA, sea salt and dust are two of the most abundant aerosols [32]. Wind-driven spray from ocean waves flings sea salt aloft. NaCl constitutes 78% of the dissolved solids in seawater [32]. Earlier studies [33,34,35] have shown that the directly observed total depolarization ratio suddenly increased in the afternoon when the sea breeze became dominant and decreased rapidly in the evening. It was also shown that nonspherical particles, dust and crystallized sea-salt particles diffused in the troposphere. The calculations showed that the diameter of crystallized NaCl particles was larger than 1 µm. Thus, we believe that these particles can serve as primers for the condensation of atmospheric moisture forming hydration shells. This is a possible way for the formation of hydrated salt microcrystals in the atmosphere. These microcrystals fall into the lower layers of the troposphere, become part of other aerosol pollution, and enter the water again, starting a new turnover cycle of salt in nature. 

In this work, we observed the dynamics of phase transitions in drying aqueous solutions to verify the earlier assumptions made about the nature of water-salt structures (giant clusters). The results of the observations allowed us to propose a scheme that fundamentally describes these dynamics in drying liquids. Additional research is needed to detail the process.

## 4. Materials and Methods 

Tap, distilled and deionized water were used in the study. The experiments were performed under laboratory conditions at *T* = 22–24 °C, *H* = 73–75%. The electrical conductivity of the solutions was measured with a MARK-603 conductometer (Russia). Only new “ApexLab” glass slides (Cat. No. 7105), glass and plastic Petri dishes (polystyrene, sterile, MiniMed, Russia) were used for our experiments without additional processing. A certain amount of the investigated water samples (specified in the text) was placed on a glass or plastic substrate and left until the free water was completely evaporated. After that, microscopic observation of the sediment dynamics was carried out for several days. The samples were examined under a Levenhuk microscope with a computer-coupled video camera Levenhuk C-1400 NG using the ToupView program. 

To test the possibility of salt ingress from the ambient air into water, the following experiment was carried out. 50 mL of distilled water from the same container was poured into two identical clean glass beakers. The end of a plastic tube connected to an aquarium compressor was immersed into one of the beakers. The input end of the tube was placed above the laboratory table in the working area. Laboratory air was passed through the water for 10 min at a rate of 72 L/h (before the experiment, the compressor was idling for 20 min to clean air paths from possible internal contaminants). Thus, 12 L of air passed through 50 mL of water during the experiment. Both beakers (control and experimental) were left on the table under room conditions, covered with a flat lid, for a week to restore the structural balance. Thus, air was passed through the water using a uniStar AIR 1000-1 aquarium compressor (2.5 W, 72 L/h) for 10 min. 

The X-ray diffraction experiment was performed on a Bruker D8 Discover X-ray diffractometer. The survey was carried out in a sliding incidence geometry (angle of incidence—3°) with a Gebel mirror and a 0.6 mm gap on the primary beam. A 2ϴ scan was recorded with a Soller gap in front of the detector.

The sample for the SIMS study was prepared as follows. A drop of deionized water was placed on a clean surface of a silicon substrate and placed in a dry-heat cabinet for evaporating water (10 min, 150 °C). Then the drug was transferred into the chamber of the device. SIMS measurements were made on the TOF.SIMS-5 system with a time-of-flight mass analyzer. We used cluster probe ions Bi^3+^ with an energy of 25 keV and a current of 1 pA in a single pulse. Separately, negative and positive secondary ions were recorded. Mass spectrometry of secondary ions (ionized sputtering products) allows surface and volumetric analysis of element concentrations to be performed. Using SIMS, a qualitative analysis was carried out with reliable identification of all elements present with a sensitivity level of Nα > 1014–1016 at/cm^3^. In addition, the image of the surface in the secondary ions (lateral resolution of 0.1–0.50 microns) was obtained. The measurements were performed in the static SIMS mode, which ensured the nondestructive character of surface analysis.

## 5. Conclusions

The results of the study and analysis of the literature enable us to make the following conclusion. Aerosol contamination of water, which is almost impossible to avoid in real life, is accompanied by the appearance in the liquid medium of “giant clusters”—micro aggregates of LCSs, each of which is formed around a salt microcrystal. Salt contained in LCSs does not dissolve, as liquid-crystal water has insufficient dissolving ability. At a certain stage of water drying, when the osmotic pressure reaches a critical level, the LCSs are eroded and “melt”. The released microcrystals of salt obtain an opportunity for further growth and “Ostwald ripening”. The methodological approach we used allows us to consistently explain the mechanism of the formation of “giant clusters” in bulk water described by other researchers. We showed a significant level of air pollution in laboratories with salt dust entering the water despite the usual precautionary measures. It was shown that the main structure-forming LCS unit is a microcrystal of sodium chloride. In our opinion, these air pollutants are part of a single salt cycle in nature, which inextricably binds water and airspace. The process of phase transitions of water and salt was traced and described qualitatively when the liquid dried on a solid substrate under normal conditions. 

## Figures and Tables

**Figure 1 ijms-20-01582-f001:**
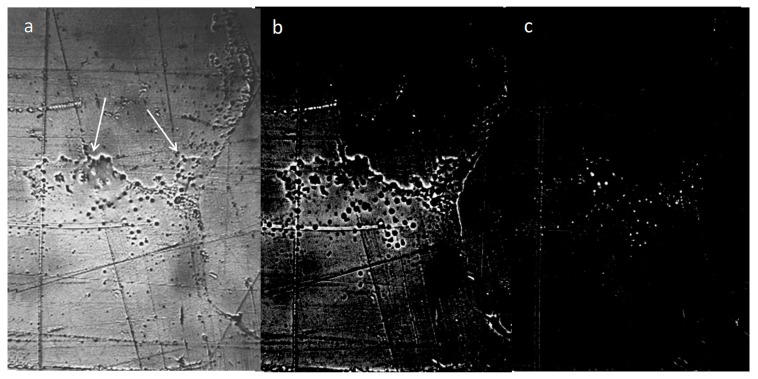
Thin film of non-evaporating water (indicated by arrows) with liquid-crystal structures. Observation on the 3rd day after evaporation of free distilled water (specific conductivity—4.1 µS/cm) from a plastic Petri dish (initial layer thickness ~ 2 mm): **a**—under ordinary lighting; **b**,**c**—in polarized light at different focal lengths (**b**—glow of liquid-crystal shells; **c**—glow of central salt microcrystals). The size of each frame—0.5 × 1.0 mm.

**Figure 2 ijms-20-01582-f002:**
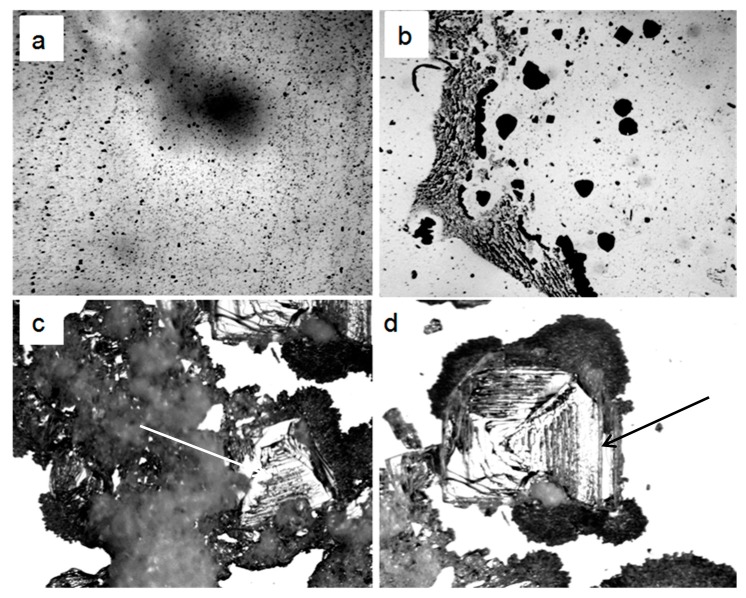
Structures formed after drying of distilled water (specific electrical conductivity is 36.5 µS/cm) in a Petri dish (layer thickness ~ 2 mm): **a**—7 days after the start of the experiment, **b**—14 days after; **c**,**d**—large crystals grown from the sediment mass (**a**) one week after scraping it with a scalpel into a single mass. Arrows point to large NaCl crystals. The width of the frames **a**,**b**—3 mm, **c**,**d**—1 mm.

**Figure 3 ijms-20-01582-f003:**
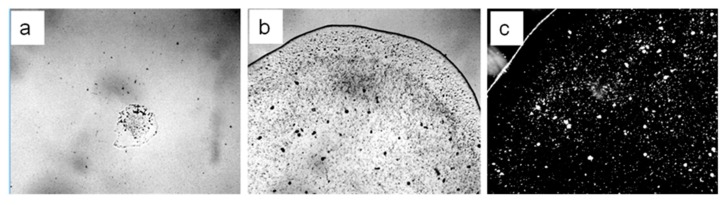
Microphotos of water droplets dried on glass: **a**—control; **b**,**c**—after passing the air (**c**—dark-field image). Frame width: **a**,**b**: 3 mm, **c**: 1 mm.

**Figure 4 ijms-20-01582-f004:**
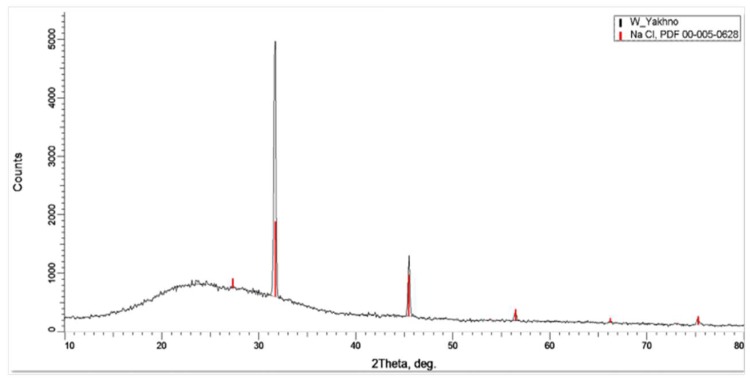
Diffraction pattern of crystals formed after evaporation of distilled water.

**Figure 5 ijms-20-01582-f005:**
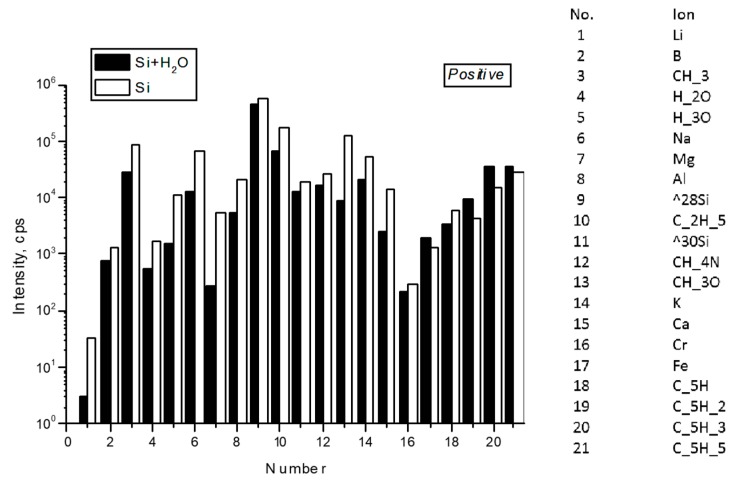
Intensity of the lines of positive secondary ions on the surface of silicon wafer inside and out of a spot of a dried drop of deionized water (light and dark bars, respectively).

**Figure 6 ijms-20-01582-f006:**
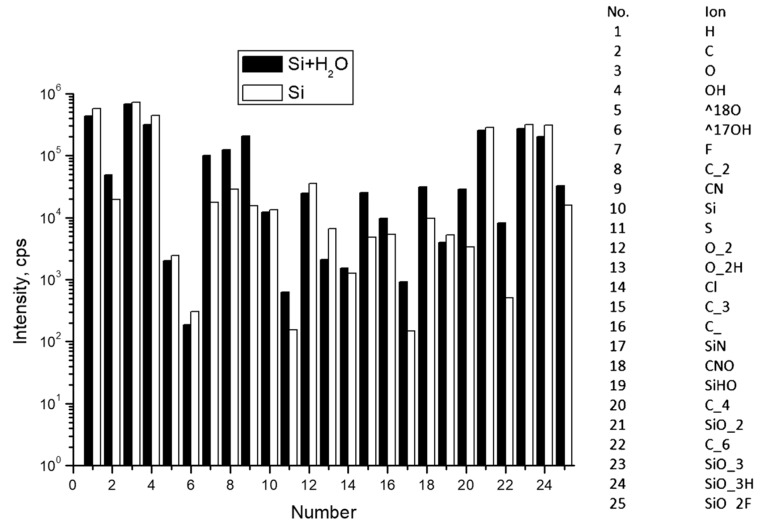
Intensity of the lines of negative secondary ions on the surface of silicon wafer inside and out of a spot of a dried drop of deionized water (light and dark bars, respectively).

**Figure 7 ijms-20-01582-f007:**
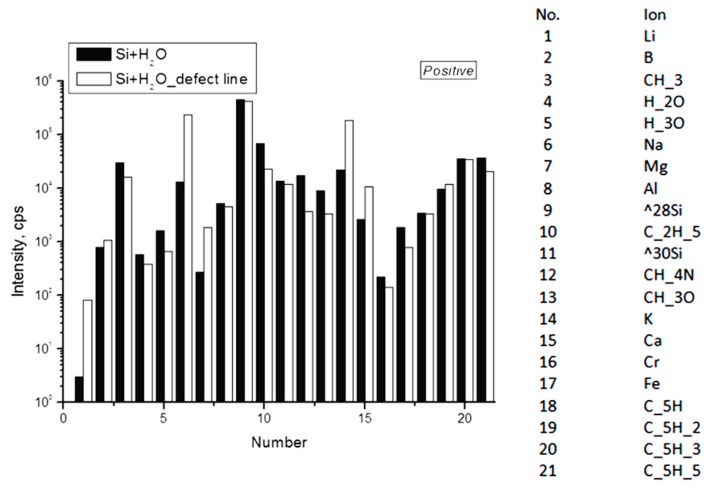
Intensity of the lines of positive secondary ions on the surface of silicon wafer after evaporation of a drop of deionized water on it: along the three-phase boundary (“defect line”) and inside the circle (light and dark bars, respectively).

**Figure 8 ijms-20-01582-f008:**
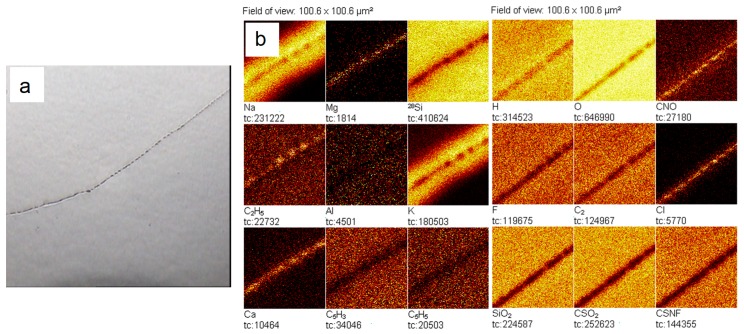
Lateral surface map in secondary ions in the area of the three-phase boundary of the dried drop according to SIMS: **a**—image of a portion of the surface of Si on which there is a fragment of the perimeter of the dried drop of water (obtained by the integrated video camera of the TOF.SIMS-5 system; frame size 1000.6 × 1000.6 µm^2^); **b**—lateral distribution of secondary ions on the Si surface, including the perimeter of the dried drop. Field of view: 100.6 × 100.6 µm^2^.

**Figure 9 ijms-20-01582-f009:**
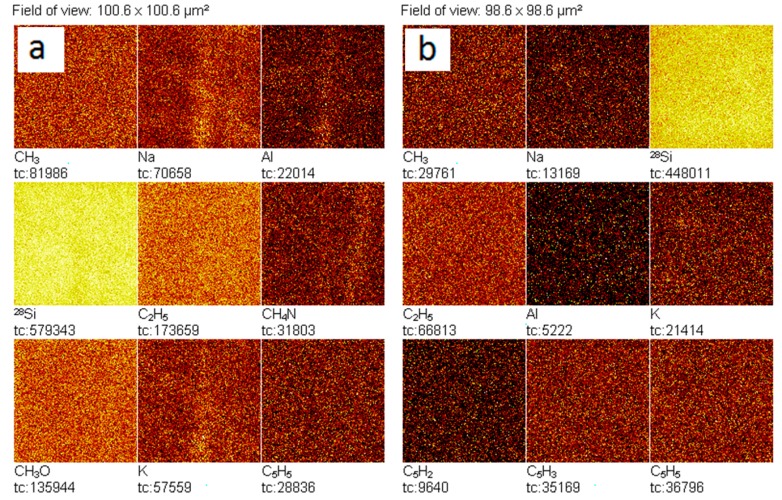
Lateral surface map in secondary ions in the area of a silicon substrate outside the zone of location of the dried drop (**a**) and inside it (**b**) according to SIMS. Field of view: **a**—100.6 × 100.6 µm^2^; **b**—98.6 × 98.6 µm^2^.

**Figure 10 ijms-20-01582-f010:**
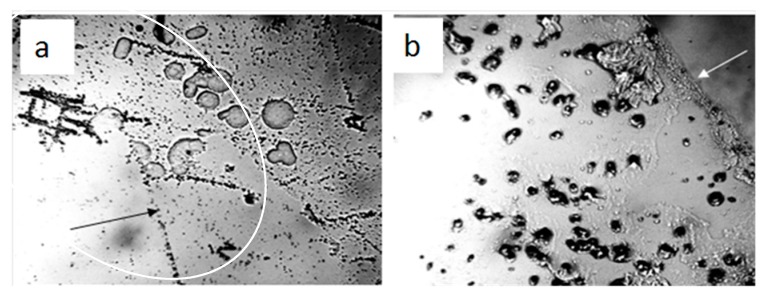
Microscopic picture of structures in tap water (1 mL, specific electrical conductivity is 550 µS/cm) dried on glass 5 days after application: **a**—melting of LCSs and merging of the formed liquid droplets (marked by an oval border); **b**—growth of larger salt crystals (dark dense structures spread over the water film). The arrows point to the edge of non-evaporating water film on a substrate. Frame width: 1 mm.

**Figure 11 ijms-20-01582-f011:**
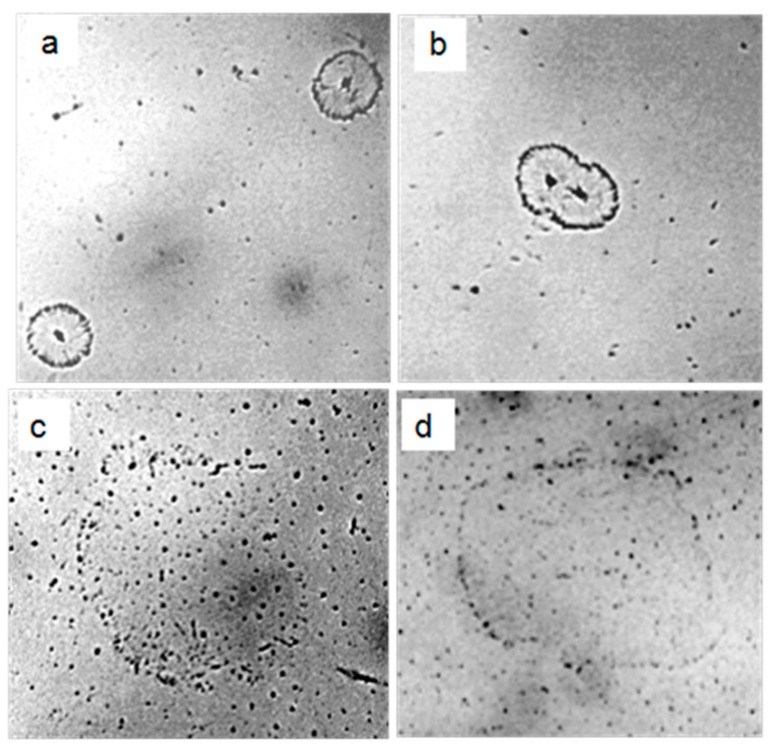
Structures on the glass after drying 1 mL of distilled water with specific electrical conductivity of 36.5 μS/cm (**a**,**b**) and 14 μS/cm (**c**,**d**). **a**,**b**—onset of salt erosion of LCSs, which is indicated by the radial striation and scalloped edges of the structures; **c**,**d**—aggregates of LCSs deposited on glass from the dried layer of liquid water. The width of each frame is 1.0 mm.

**Figure 12 ijms-20-01582-f012:**
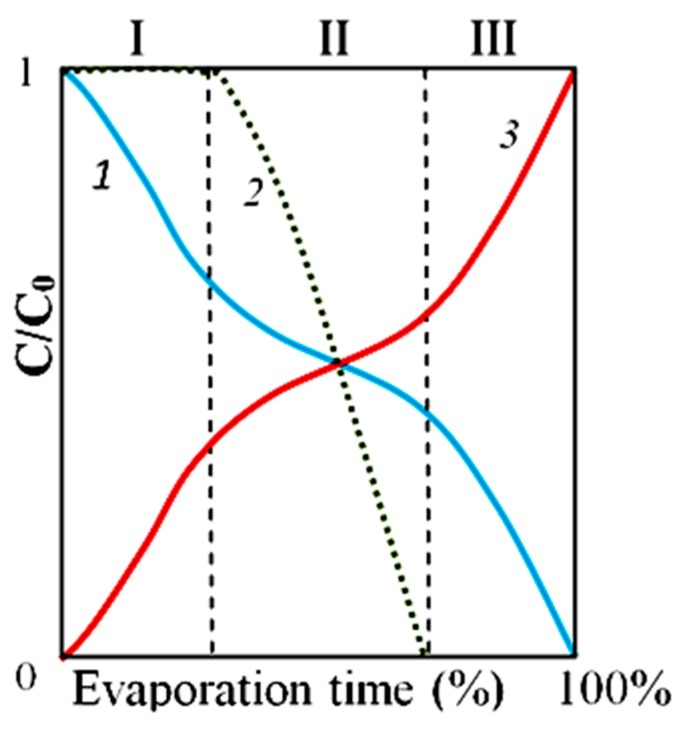
Schematic diagram of the dynamics of phase transitions in water containing LCSs with NaCl microcrystals as a “seed” during its evaporation from a solid substrate in the “Relative concentration–Evaporation time (%)” coordinates. Stage I—evaporation of free water and increase in osmotic pressure; Stage II—phase transition of liquid-crystal water into free water and reducing relative rate of evaporation; Stage III—growth of NaCl crystals. 1—relative concentration of free water; 2—relative concentration of liquid-crystal water; 3—relative salt concentration.

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
