# Peer review of "Giant Water Clusters: Where Are They From?"

_ijms, 2019, doi:10.3390/ijms20071582_

Round 1
Reviewer 1 Report
My comments and changes to the manuscript can be seen in the word format of this paper.
As you notice, the paper must undergo extension revision. As an example, I have made changes to the methods, some parts of the results and the discussion sections.
Discussion section is repetition of the results.

Author Response
Please find our response in the attached file

Reviewer 2 Report
The authors have taken the opportunity to improve the paper through the review process. It now seems appropriate that the paper should be published such that the wider scientific community can draw its conclusions.
Author Response
Dear the Reviewer_2,
Many thanks for recommending our manuscript for publication. We hope to continue our work and to involve independent researchers in the search for truth.
Respectfully,
Tatiana & Vladimir Yakhno, Mikhail Drozdov.
Round 2
Reviewer 1 Report
Authors have worked hard to improved this manuscript. It can be published.
This manuscript is a resubmission of an earlier submission. The following is a list of the peer review reports and author responses from that submission.
Round 1
Reviewer 1 Report
I have provided detail comments in the manuscript. Please go through each and rework the manuscript.
In general, your work provide important rationale for environmental pollution, but this has not come out as written.
More detains are needed in the Methods section.
Figure quality must be improved. I see lots of structures in the figure without knowing what they are. Improve legends.
Discussion section needs to be expanded. Please do not start discussion with yous earlier work and say that the present work simply reaffirms that. Not a good strategy.
ENGLISH: Make sure that your paragraphs have some continuity. One paragraph must lead to the mext.

Reviewer 2 Report
This paper is essentially a sister paper to the paper “A Study of the Structural Organization of Water and Aqueous Solutions by Means of Optical Microscopy” written by two of the three authors and recently accepted for publication in MDPI:crystals. The two papers reference each other. The previously published paper uses optical microscopy to demonstrate the presence of microscopic spheres of water with a diameter of the order of 10 µm in a liquid crystalline state with a microcrystal of the order of 3 µm diameter at its centre. In the paper under review these crystals were isolated, e.g. by droplet evaporation and X ray diffraction and secondary ion mass spectrometry used to identify these crystals as being NaCl. Microscope pictures are presented of the residue left after droplet evaporation as evidence of the structures present in liquid water. Previous work is cited to show that even in ultrapure water, there is will be sufficient NaCl present to catalyse the creation of the giant water clusters that are seen. Previous work is also used to show that the ambient atmosphere contains sufficient suspended NaCl crystals to explain the NaCl concentrations seen in different grades of water purity.
The X-ray diffraction data in the paper does appear to provide clear, and perhaps not unexpected, confirmation that the majority solute that crystallises out from the distilled water is NaCl. There are however significant issues with the Secondary Ion Mass Spectrometry. The nature of the structures that remain when tap water evaporates on a glass surface is ambiguous and more information is needed before the results can be taken as an indication of the presence of LCS within the water. I therefore think that the experimental data in the paper is insufficient to justify the phase transition diagram in Fig 9, although it can be argued that it is supported by data published elsewhere.
Lines 109-112. The presence of significant, and similar levels of a large number of elements in the control and the experiment would appear to make these data unusable as a way of determining the content of a drop of water on its surface. The data are consistent with there being significant levels of contamination of the silicon surface and the apparent increase in carbon and sulphur could well be due to sample to sample variation in this contamination. I believe that these results are however tangential to the primary conclusions of the paper which is to show the presence of NaCl. Indeed Fig 4 shows an apparent drop in the quantity of Na, which contradicts the main conclusion of the paper.
Lines 116-120 The line in Fig 6 is supposedly at the perimeter of the water droplet. However there is no clear difference between the signal ‘inside’ and ‘outside’ the water droplet making it very unclear how it might be legitimate to use one of them (the upper?) as a reference for the concentration in the central zone of the spot.
Lines 129-137 and Fig 7. While the text is a reasonably clear description of what is believed to be happening, the supporting evidence from Fig 7 is far from clear. For example, It is not clear how Fig7a shows merging of droplets and Fig7b is supposed to show the growth of larger salt crystals, but there is no clear ‘before’ comparison.
Line 151-153 and Fig 9. I have a suspicion that the labels for lines 2 and 3 are the wrong way around, as the relative concentrations of free and LC-water should sum to 1.0.
Lines 142-145 and Fig 8. Without further information, the figures do not provide clear evidence for LCS. When the water evaporates, it is likely that in the final stages there will be a small number of microscopic droplets remaining around particles (crystals) that had settled on the surface earlier in the evaporation process. When they reduce to a certain size the solutes will become saturated and start crystallising out at the edge of the droplet. This would appear to be a simple non-LCS alternative explanation for Fig 8a and b. More information is required to distinguish between the LCS and non LCS interpretation of these results. The same is true for Fig 8c and d.
Discussion
Line 166. The sentence “The content… water is 78%” is incorrect as it stands. I think this may be saying that NaCl constitutes 78% of the dissolved solids in seawater. Reference needed.
Line 176 The figures of 80% and 90% require a reference
Line 180 Reference 34 would suggest that the temperature at 10km is nearer to -60°C, not 1.5°C.
Line 180-181. While it may be true, the relevance of the formation of NaCl+2H20 crystalline hybrids at low temperatures to the discussion and conclusions of the paper is far from clear.
Line 177-189 These paragraphs would appear to be suggesting that seawater droplets are drawn up into the upper atmosphere and as they cool the other salts crystallise out leaving relatively pure NaCl solution which then form microcrystals, presumably as the droplet continues to evaporate. This seems an unlikely, and unnecessary mechanism for the elimination of salts other than NaCl, It would seem more likely that interaction between the wind and sea at sea level is responsible for the salt crystals that are suspended in the atmosphere, much as is suggested by lines 168-170.
Materials and methods
I am aware of experiments which show that the way in which silicon and glass surfaces are cleaned can have a dramatic effect on the crystallisation of solutes out of water as drops evaporate (e,g, cleaning with solvents v. cleaning with strong acids). The cleaning technique used should be included in the methods section.
Conclusions
My previous observations and comments mean that the only conclusion that I believe can be drawn from the data in the paper is that water, even supposed ultrapure water, contains NaCl.